# Ke ala i ka Mauliola: Native Hawaiian Youth Experiences with Historical Trauma

**DOI:** 10.3390/ijerph191912564

**Published:** 2022-10-01

**Authors:** Lorinda Riley, Anamalia Suʻesuʻe, Kristina Hulama, Scott Kaua Neumann, Jane Chung-Do

**Affiliations:** 1Thompson School of Social Work and Public Health, Office of Public Health Studies, University of Hawaiʻi Mānoa, Honolulu, HI 96822, USA; 2Department of Psychology, College of Social Sciences, University of Hawaiʻi Mānoa, Honolulu, HI 96822, USA; 3Thompson School of Social Work and Public Health, Social Work, University of Hawaiʻi Mānoa, Honolulu, HI 96822, USA; 4Humanities Division, University of Hawaiʻi West Oʻahu, Kapolei, HI 96707, USA

**Keywords:** indigenous people, historical trauma, wellbeing, Native Hawaiian

## Abstract

Native Hawaiians (NH), like other Indigenous peoples, continue to experience the subversive impacts of colonization. The traumatic effects of colonization, especially the forced relocation from land that sustained their life and health, have led to complex, interconnected health disparities seen today. NHs have described a collective feeling of kaumaha (heavy, oppressive sadness) resulting from mass land dispossession, overthrow of the Hawaiian Kingdom, cultural loss, and early loss of loved ones. Although historical trauma is linked to high rates of substance misuse, depression, suicidality, and other mental health disparities in American Indian populations. However, the link between NH historical trauma and health disparities among NHs has been less explored. This qualitative study used Indigenous talk story interviews with 34 NH ʻōpio (youth) and ka lawelawe (service providers) to explore how NH ʻōpio understand and experience historical trauma. Eight themes and 35 sub-themes were identified covering individual, community, and systemic domains representing the first step in addressing NH historical trauma.

## 1. Introduction

Native Hawaiians (NH), like other Indigenous peoples, continue to experience the subversive impacts of colonization. The traumatic effects of colonization, especially the forced relocation from land that sustained their life and health, have led to complex, interconnected health disparities seen today. NHs have described a collective feeling of kaumaha (heavy, oppressive sadness) resulting from collective land dispossession, overthrow of the Hawaiian Kingdom, cultural loss, and early loss of loved ones [1,2]. Although historical trauma is linked to high rates of substance misuse, depression, suicidality, other and mental health disparities in American Indian populations, little empirical work has been done to understand how Native Hawaiians experience historical trauma.

Historical trauma is defined as the cumulative, emotional, and psychological wounding over one’s lifespan and across generations. Globally, historical trauma is present in survivors of colonization, genocide, and dispossession, and it is linked to high rates of substance use [3,4], depression [5], suicidality [6], other and mental health disparities [7]. Historical trauma is rooted in a history of colonization and is compounded in the present day through persistent racism, microaggressions, and epigenetic expressions [8] reflecting environmental trauma in genetic makeup. Prior studies indicate that NHs conceptualize historical trauma as kaumaha or a heavy, oppressive feeling of sadness or despair due to loss of land, governance, health, education, culture, language, and early loss of family members among NHs [1,2].

NHs are one of the fastest growing populations in Hawaiʻi, yet they disproportionately suffer from physical, mental, and social ills. NHs today exhibit disparate rates of chronic disease, mortality from cancer [9,10], and obesity [11] with NHs being twice as likely to be obese than Caucasians in Hawaiʻi [12]. Mental illness [13,14], including depression [15], drug use [16], and other forms of trauma are more prevalent among NHs compared to other ethnic groups in Hawaiʻi [17]. Additionally, NHs exhibit symptoms of post-traumatic stress disorder, including anxiety, detachment and anger, which are directly linked to historical trauma [17,18]. Moreover, NHs have the highest prevalence of adverse childhood experiences (ACE [19]), which are associated with behavioral and health problems [20] such as depression [21], suicidality [22], drug misuse [23], and incarceration [24].

Because NH ʻōpio (youth) make up a large portion with over 40% of the State’s population, understanding the impacts of historical trauma is particularly important. In 2012, NHs made up approximately 30% of the juvenile arrests, yet made up 56.9% (53.1% of boys and 69.1% of girls) of the population in the Hawaiʻi Youth Correctional Facility (HYCF) [25]. In 2017, the degree was smaller, however, NHs were still disproportionately represented (44.5% of detained ʻōpio) in the juvenile justice system [26]. NH ʻōpio are also overrepresented in the Hawaiʻi Child Welfare System with 45% of the ʻōpio in foster care [27], despite being only 26% of the minor population.

According to a 2019 report on Hawaiʻi public schools, Pacific Islanders, which includes NH, made up 26% of the student body, but made up 66% of ʻōpio suspended. Pacific Islanders in Hawaiʻi are arrested at higher rates than any other state in the US [28]. The penal focused school disciplinary system has perpetuated the school-to-prison pipeline and increased the number of contacts minority ʻōpio have with the juvenile justice system. Frazier and Cochran found that contact with the juvenile justice system increases the chances of adult imprisonment [29] three-fold and time spent in custody was the strongest predictor of future imprisonment [30].

Despite this conceptual link, little research has been undertaken to explore how NHs experience historical trauma and the impact that this may have on health and wellbeing, especially for youth. In particular, how ʻōpio experience and understand historical trauma can help identify programs and services that can be implemented to mitigate some of the kaumaha of future generations. This study addresses this gap by conducting a qualitative analysis of Indigenous talk stories with Native Hawaiian ʻōpio and ka lawelawe (service providers/legislators). If left unaddressed a significant part of the future generation will be at risk of failing to reach their full potential.

### 1.1. Background

Conceptual models are useful in representing a system. Although there are several conceptual models of historical trauma [31,32,33], we used Sotero’s Conceptual Model of Historical Trauma [34] because it was developed for Indigenous communities and incorporates the nuances of colonization. Sotero’s model is grounded in 3 interconnected theories that link intergenerational disease with: (1) environmental stress stemming from psychosocial theory; (2) power imbalances arising out of political/economic theory; and (3) multi-level dynamics of social ecologies emanating from ecological systems theory.

Under Sotero’s model the first-generation experiences of the primary trauma of colonization, which takes the form of cultural degradation, physical violence, displacement, loss of resources, and loss of political sovereignty referred to as the “mass trauma experience”. [34] These experiences are tempered by resilience and protective factors, which tend to diminish with each subsequent colonized generation. Individuals in this generation can experience psychological trauma (e.g., nutritional stress and biochemical abnormalities); social responses (e.g., increased suicide, substance use, and unemployment); and psychological responses (e.g., post-traumatic stress and depression) [34].

The primary generation then transmits this trauma intergenerationally to later generations, creating lasting impacts (both distal and proximate) on health and well-being [34]. While later generations may no longer experience all elements of direct colonial trauma they are still influenced by the prior generation’s historical trauma as well as present day discrimination and subversive colonial trauma (e.g., lack of government accountability, inability to access sacred lands, reduced population vis-a-vis the colonial power, etc.) [3,34,35]. These later generations carry, not only the individual’s current trauma, but the culmination of the prior generation’s trauma and thus may be subjected to higher composite levels of historical trauma with less access to protective factors.

### 1.2. Indigenous Historical Trauma

Over the past few decades, research around the conceptualization and measurement of American Indian (AI) Historical Trauma has grown. Much of the groundwork on Indigenous historical trauma was laid by Brave Heart who outlined a number of symptoms associated with historical trauma [3] as well as its relationship with substance abuse [35]. In a culturally syntonic, psychoeducational intervention, Brave Heart investigated the effectiveness of the grief and trauma resolution intervention among 45 Lakota service providers and community leaders. Findings indicated that the intervention increased both awareness of historical trauma and grief affects among participants, with grief responses decreasing through sharing the resolution process with other Lakota. Increased pride was also present post-intervention [36].

Building on this work, scales measuring historical trauma symptoms have deepened the understanding of how historical losses relate to daily thoughts and emotions. The development of a scale to measure historical trauma provided an opportunity to explore the relationship of historical trauma with an array of behaviors or symptoms. The Historical Loss Scale (HLS), developed among Native American Plains tribes, measures the frequency of thoughts of 12 types of losses, including loss of land, language, or people through early death [2]. Whereas the Historical Loss Associated Symptom Scale measures the symptoms that may be associated with such losses such as sadness, anger, and anxiety [2].

Among American Indian adolescents, thoughts of historical loss were associated with emotional distress and increased stress [37]. Behavioral outcomes including substance dependency have also been associated with thoughts of historical loss. [3] Ehlers et al. found that American Indian participants with diagnoses of substance dependence had significantly higher scores on both the HLS and Historical Loss Associated Symptom Scale [4] (HLASS). Another study among American Indian college students found that strong ethnic identification was positively associated with historical loss thinking [5]. Identifying the different effects associated with historical trauma further demonstrates its complexity.

Along with a range of symptoms or manifestations, another key element of AI historical trauma is intergenerational transmission. [34] The pain of historical trauma reaches across the lifespan of individuals and throughout generations of a community. Both older and younger generations can be affected and studies have found that Indigenous adolescent participants suffer from thoughts about historical loss at similar rates to older participants [4,38]. These findings suggest that younger generations are still impacted by historical trauma, even though they have not directly experienced certain traumatic events (i.e., boarding schools) furthering evidence of the intergenerational transmission component of historical trauma.

In a systematic review of studies investigating Indigenous historical trauma and adverse health outcomes, Gone et al. found three categories of focus among 32 studies: historical loss, residential school ancestry, and “other,” which included an amalgamation of studies [39]. Methods, participants, findings, and interpretations varied greatly making Indigenous historical trauma difficult to synthesize and suggesting that more work needs to be done in this area [39]. New research in epigenetics has found that trauma is expressed through genetic mutations adding biological contributions to the intergenerational transmission of historical trauma [8].

### 1.3. Historical Trauma among Native Hawaiians

NHs experience historical trauma in similar, though not identical, ways as American Indians. Significant work has been done articulated the impact that colonization has had on NHs. Cook et al. discussed the myriad of dramatic and sudden change since western contact [40] that has had a traumatological impact on NHs, especially in relation to spirituality and gender roles. Kaholokula also suggested that “U.S. colonialism in Hawaiʻi and its acculturation process have led to adverse physical, psychological, and social consequences” for NHs. Rezentes conceptualized historical trauma as kaumaha or a heavy feeling of loss, sadness, or despair [41]. Crabbe called this historically connected trauma hōʻinoʻino or broken spirit, a type of collective depression [42].

Despite the fact that there has been theoretical discussion of NH historical trauma, only two empirical studies have applied the validated HLS to NHs. The first, explored NH historical trauma in relation to substance use [43]. Researchers applied the HLS scale to NH college students and found that the frequencies that NH thought about historical loss was less than among AI, with NHs reporting thoughts of historical loss “yearly or at special occasions” verses “daily” or “several times a day” for AI [43]. In addition, they found that perceived discrimination mediated the effects of historical trauma on substance use and mean levels of historical traumatic events were lower among NH compared to AI. The second study was qualitative and found that NH described historical loss in ways that were distinct from AI, including militarization, Christianity, the overthrow of the Kingdom, and the importance of mahū perspectives [44]. Currently, there is no way to measure the historical trauma that accounts for these differences placing NHs at a disadvantage because they are unable to definitively link negative health outcomes to the underlying cause of historical trauma.

In this paper we explored whether and how NH ʻōpio (aged 15–24) understood and experienced historical trauma. After providing background on the origins of historical trauma, we expounded on the application of historical trauma theory to Indigenous communities, specifically Native Hawaiians. Using Indigenous talk story method we conducted 34 talk stories of NH ʻōpio and lawelawe before qualitatively analyzing the data. The data indicated 8 interconnected themes and 35 sub-themes. This research paints a picture of the very real and damaging impact that Hawaiʻi’s legacy of colonization has exacted and continues to perpetuate on NHs. We end our article with a discussion of valuable next steps to address historical trauma to end the detrimental cycles that prevent NHs from achieving their full potential in Hawaiʻi.

## 2. Method

### 2.1. Position Statement

As Indigenous qualitative scholars, the main research team values the expression of positionality and the influence that our worldview and experiences have on the inquiry process and interpretation of findings. Most of the researchers who participated in this study are Indigenous and work at an institution that brands itself as an Indigenous serving institution of higher education. One researcher serving as a mentor throughout this process is non-Indigenous, but is an ally who has deep connections within the local Hawaiian community. Our positionality encourages us to reflect deeply on our place within the academy and also within our respective communities. Engaging community partners at all levels throughout the process, and especially in the conceptualization of the research design was an important element of our process. Moreover, recognizing our relative privilege, the larger project encouraged us to incorporate reciprocal elements, including developing materials for community dissemination, as an integral part of our work. Our privileged position motivates each of us to continue to work to improve Indigenous well-being in our communities in ways that uplift Indigenous values. This work is a representation of our values and commitment to our respective communities.

### 2.2. Indigenous Qualitative Methods

In order to better understand how NH ʻōpio experience and understand historical trauma, Indigenous methods were used in the design and implementation of this project. Although Indigenous methodology covers a broad array of methods, there are some fundamental principles that must be considered [45]. First, pilina or relationality is critical. We, thus, developed strong relationships with community partners to inform the utility of the project and collaborate with. Second, is relevance, which speaks to the project being one that the community finds important. Third, reciprocity or the idea that research should not only benefit academia, but also benefit the community. Having a strong relationship with the community aids in the process of developing a project that is relevant and mutually beneficial.

This project used an Indigenous community-engaged research design centered on reciprocity and collaboration. In the conceptualization process, we connected with a broad array of Native Hawaiian community members to gain a better understanding of community needs and interests. The research team identified several community partners serving at-risk NH ʻōpio including incarcerated ʻōpio, to collaborate with, including Kawailoa Youth and Family Wellness Center (KYFWC), Hawaiʻi Youth Correctional Facility (HYCF), Residential Youth Services and Empowerment (RYSE), and Adult Friends for Youth (AFY). Initial meetings with community partners discussed study goals, procedures, recruitment, ethics, and dissemination. These community collaborators were engaged from idea development to dissemination and hopefully beyond as we continue this work. As a result, the data collection focused on two populations: (1) Native Hawaiian ʻōpio (aged 15–24) and (2) lawelawe and policy makers that worked on NH ʻōpio issues.

Because our community partners identified trust as an obstacle to data collection, the research design incorporated the community partners as co-collectors of ʻōpio perspectives and stories, unless the ʻōpio wanted to do the talk story alone. All community partners who engaged in this project obtained CITI Training and were trained on our research protocol and methods. During recruitment, community partners identified eligible ʻōpio in their programs and coordinated with the research team to set up virtual or in-person talk stories, based on ʻōpio preference. Due to COVID-19 protocols, the first part of this research project was limited to virtual talk stories.

### 2.3. Talk Story Sessions

Participants included 19 NH ʻōpio, 13 lawelawe, and 2 Hawaiʻi legislators. ʻŌpio all identified as NH between the ages of 15 to 24. The initial set of ʻōpio (*n* = 10), interacted with the Juvenile Justice System to some degree (e.g., arrests or incarceration). Later, we expanded our population to mirror the symptoms identified by Whitbeck in the HLS and HLASS (i.e., poverty, periodic sadness, anger, anxiety, distrust of the intentions of those in power, used controlled substances, or had family members with substance dependency). See Demographic Table for full description (Table 1).

### 2.4. Analysis

After the talk story sessions, the sessions were transcribed verbatim and uploaded into Atlas-ti for analysis. The transcribed sessions were independently coded in Atlas-ti utilizing a redundant coding scheme. Conventional inductive content coding was utilized to identify words, statements, or phrases related to historical trauma. We utilized a rigorous process, including investigator triangulation to independently code and meeting weekly to discuss the findings. Coders were encouraged to write memos in Atlas-ti to clarify codes and help identify discussion topics for our meetings. Coders agreed upon an initial codebook and re-coded the talk stories using selective coding.

Codes were sorted into categories and linked to create categories. In developing categories and themes the researchers referred to the Ola Triangle [46], a conceptual framework of NH health and wellbeing, emanating from the Lokahi Triangle [47], to ensure internal consistency with NH worldviews. Themes and sub-themes then emerged as the categories were grouped. Rigor was also ensured through the triangulation of data sources covering talk story sessions with NH ʻōpio as well as lawelawe. Each of the themes were discussed by both NH ʻōpio and lawelawe indicating consistency among participants. Additionally, because participants had the option of selecting either an individual or group talk story there was methodological triangulation with the inclusion of four group sessions.

Throughout the manuscript, we opted to maintain the integrity of the participants’ speech patterns. Many Native Hawaiians and Hawaiʻi residents speak Hawaiian pidgin, a creole developed through the patchwork of people that immigration or were indigenous to Hawaiʻi. We have attempted to stay as authentic as possible in how our participants chose to share their thoughts and have provided translations if necessary. Because language is so important to capturing the NH experience with historical trauma, two fluent Hawaiian language speakers were consulted, one of whom is an author to this paper, to ensure conceptual fluidity of themes.

## 3. Results

A total of 8 themes and 35 sub-themes were identified across the 34 talk stories. All participants discussed experiencing or witnessing trauma in the present and trace that trauma to collective, historical, or colonial trauma. However, a few participants (particularly ʻōpio) struggled to articulate their present-day trauma in terms of colonial trauma. Nonetheless, these ʻōpio were able to articulate negative social, cultural, and lifestyle changes that they attributed to colonization. See Theme Table for a list of all themes and sub-themes (Table 2).

### 3.1. Theme 1: Peʻa ka Lima i ke Kua (Emotions)

ʻŌpio and lawelawe described strong emotions relating to Native Hawaiian history or historical trauma as well as present-day circumstances for Native Hawaiians. Peʻa ka lima i ke kua means to cross the hands behind the back and is a physical expression of the strong emotions that ʻōpio shared (i.e., anger, grief, shame, or hopelessness). Theme 1 Naʻau (Emotions) covers the strong emotions ʻōpio experienced. Both negative and positive emotions were discussed, however, ʻōpio focused more on their experiences with negative emotions. Six sub-themes were identified including Pain/Sadness, Anger, Loss of control, Hopelessness, Pride and Anxiety/Fear.

#### 3.1.1. Ke Kaumaha (Pain/Sadness)

ʻŌpio shared feelings of pain, grief, or sadness when talking about changes in Hawaiʻi over time. Some ʻōpio attributed these changes to the impact of settlers and colonization in Hawaiʻi, such as the decimation of the Native Hawaiian population and illegal overthrow of the Hawaiian Kingdom were sources of sadness for ʻōpio. One ʻōpio shared their feelings of past and ongoing injustices in Hawaiʻi, “And just knowing how Hawaiʻi was overthrown is heart-breaking. And how we have so many things that were taken from us. And we’re still continuing to lose today. It’s … it’s really not fair”. However, some ʻōpio did not connect their feelings of pain or sadness to Hawaiian historical events, but rather associated them with their present circumstances.

Lawelawe also observed a deep sadness or grief among Native Hawaiian families they worked with and emphasized an intergenerational element to this sadness. Many lawelawe were Native Hawaiian themself and were able to recognize the pain and sadness in their own families and its impact on their work, “I went through a lot of tears because it brought back my family’s grief. From my father, you know … alcoholism and abuse and all of that you know would have [stemmed] from his father. So I can see [the] historical trauma that happened to my family”. Lawelawe recounted the difficulty ʻōpio had in understanding the reason for their sadness or its connection to historical trauma. One lawelawe described an interaction with a Native Hawaiian child who had been in and out of foster homes, “I remember him just breaking down and just saying to me, ‘Do you know why I’m sad?’” Lawelawe and some ʻōpio described pain, grief, and sadness across multiple levels, most often in family histories.

#### 3.1.2. Ka Huhū (Anger)

Several ʻōpio shared experiences with anger issues and trying to find ways to mitigate their anger. One ʻōpio spoke about school counselors providing “squish balls” to decrease their anger. However, these interventions only added to their frustration, “[I]t would make me more mad when they would try to help me and it doesn’t work. Like I get mad at the smallest stuff”. Several ʻōpio stated that they were going to therapy for their anger issues.

Some ʻōpio described punching, scratching themselves, and other types of self-harm in order to cope with their anger. Several ʻōpio shared “crying” out of frustration. Other ʻōpio mentioned yelling or staying quiet when they were frustrated depending on the situation. One ʻōpio who could not adequately diffuse their anger said, “*I can’t talk about it cuz I just whack people*”. Professional participants described a loss of control in the lives of ʻōpio, which sometimes presented as “[acting] out in different ways”. One lawelawe described “brutal” fights, but when they asked ʻōpio what made them fight the response would inevitably be, “I don’t know”. One lawelawe attributed these behaviors to ʻōpio not “[going] through the developmental stages” due to neglect or severe abuse, hindering their social and emotional development. Other lawelawe saw the immediate connection of these behaviors to “deep grief or trauma or kaumaha” not only within the ʻōpio, but their families as well.

Similar to the emotions of pain and sadness, an understanding of why they were angry was often difficult to pinpoint for ʻōpio. As one lawelawe shared, even adults could not articulate the reasons for their anger or violence and would sometimes describe it as a “flying rage” with “nothing” else going on in their mind.

#### 3.1.3. Ka Mānewanewa (Loss of Control)

ʻŌpio expressed feeling a loss of control at the individual and communal level. At the individual level one ʻōpio shared, “I don’t like going into detail into, like, stuff that I have no control over”. ʻōpio described feeling like they did not have control over their present situation saying that right now things were just “chaos”. At the communal level, ʻōpio expressed feeling like “we [Hawaiians] barely have control over anything”. This was reiterated by another ʻōpio who shared, “Our people can’t do anything, we’ve learned that”.

Some ʻōpio connected the feeling of losing control to negative behaviors. “Lot of people who feel the same way is like, a lot of people on the street[.] [T]hey feel so out of control about their situation [ ] and that’s why drugs is a big problem here. Coz it’s very numbing”. Both ʻōpio and lawelawe articulated a desire to increase control over decisions made in Hawaiʻi. This increased control was connected to perceived increase in happiness. Thus, participants were aware of the diminished power that NHs had compared to others in Hawaiʻi.

#### 3.1.4. Pau ka Pono (Hopelessness)

Feelings of hopelessness were described by ʻōpio especially when talking about their experiences in unsupportive systems. Several ʻōpio adopted an almost defeatist attitude when discussing their current situation. After reflecting on the displacement of Hawaiians and a decrease of “peace” in Hawaiʻi, one ʻōpio would finish their statements by saying, “it is what it is”. ʻŌpio, particularly those currently incarcerated, would also describe this feeling as not knowing how to better their situation and feeling like there is “nothing I can do about it”.

Often described hopelessness was a sense of worthlessness or low self-worth. Professional participants spoke of cycles that compounded feelings of hopelessness and low self-worth for ʻōpio and their ʻohana (family). As one professional participant shared, “So they have no motivation to really want to do better because this is just what the cycle is right?” As with other emotions like pain and sadness, hopelessness was also seen to be passed down through the generations preventing ʻōpio and families from healing.

#### 3.1.5. Ka Hopohopoalulu (Anxiety/Fear)

Some ʻōpio shared struggles with anxiety or stress. One ʻōpio shared how their anxiety made it difficult to express themselves when distressed, “I’m an overthinker and my overthinking, is like, really bad. And I have, like, anxiety and stress and stuff, so when I’m, like, mad or something, I don’t say anything”. Other ʻōpio recalled how they would feel worried, but the presence of their ʻaumakua [family diety often taking the same of an animal] or family would help ease some of these feelings, “I used to be scared and worried. [But now] I feel like I’m moving and protected. No matter where I go. I’m being watched, like, in a good way”.

Lawelawe also observed a sense of fear among ʻōpio and parents, often in regard to retaliation or punishment from government entities resulting in further loss for families. Social workers or police were seen as “triggers” for families “Because of so many children being taken” and subsequently lost to the system.

Some lawelawe connected feelings of fear with anger. One professional believed anger as a product of “constantly just reacting from fear”. Another lawelawe also acknowledged fear as the root of anger. In order to resolve issues with both anger and fear they outlined a path to empowerment saying, “anger is generally based on fear. And what can we do to make you less fearful and re-empower you? Cuz Hawaiians were empowered. And then, they were disempowered. So now we need to re-empower”.

#### 3.1.6. Ka Haʻakei (Pride)

Learning about or practicing their Hawaiian culture often brought up feelings of pride for ʻōpio. One ʻōpio shared that while they were learning about Hawaiian culture they felt, “the Hawaiian blood in me was coming out”. Pride was also a driver to participate in cultural programs as some ʻōpio felt that they were “Hawaiian to the max”. They shared a particular affinity for programs that offered a way to give back to the ʻāina (land), which many ʻōpio and lawelawe identified as a Hawaiian value.

Several ʻōpio explicitly stated “I’m proud to be Hawaiian”. The pride ʻōpio shared would often be accompanied by a drive to protect or defend their culture. One ʻōpio remarked, “I’m proud to be Hawaiian. I’m proud to live in Hawaii. It’s important to me because I would like, I would hate to see the cultural practices disappear,” while another stated, “I strongly stand for Hawaiian sovereignty because that is where my pride lies, yeah?” Even though these ʻōpio also recognized the trauma accompanied with their history, they maintained a sense of resiliency through their pride.

Some professionals observed a “sense of pride growing” among recent generations as opposed to shame and stigma from years before. Other professionals connected this shift in pride to movements such as Mauna Kea, because “Mauna Kea has brought out a lot of cultural pride”. Professionals who spoke of growing pride throughout the community noted its importance for well-being, especially among ʻōpio.

### 3.2. Theme 2: ʻAuana i ke Kula ʻo Kaupeʻa (Escapism)

The American Psychological Association defines escapism as a behavior to distract oneself from life problems [48,49]. Several strategies were discussed during talk stories as means to escape past or present pain. The second theme of ʻAuana i ke kula ʻo Kaupeʻa is drawn from stories of a plain in ʻEwa where spirits wander aimlessly. This theme outlines modes of escapism shared by participants. Four sub-themes were identified including substance use, technology as a distraction, avoidance, and consumerism.

#### 3.2.1. ʻAi Lāʻau ʻino (Substance Use)

Both ʻōpio and professional participants recognized the prevalence of substance use in Hawaiʻi. Most ʻōpio saw drugs as a new issue affecting Hawaiians today whereas substance use was not a problem before non-Hawaiians arrived. One ʻōpio reflected, “I think, like, if people, like, from the mainland never come over here [ ] we could have been living peacefully. Like without getting in trouble. [ ] Using drugs, people wouldn’t be using as much drugs”. The majority of lawelawe described substance abuse as a widespread issue for NHss most often citing problems with alcohol and methamphetamine.

ʻŌpio and professionals also spoke on substances’ role to escape negative emotions. Several ʻōpio shared their experiences with addiction and the difficulties they had with achieving and maintaining sobriety. However, ʻōpio did not always connect their substance use to their feelings about Native Hawaiian history or historical trauma. More often when ʻōpio felt angry or frustrated at their present situation they would “end up drinking” or “Smoke it out”.

In contrast, lawelawe recognized substance use, as well as other detrimental behaviors such as domestic violence, as symptoms of historical trauma rather than a solely present-day issue. One lawelawe shared, “I consider those things the results of kaumaha or historical trauma as major factors that have passed down to the generations, but [youth] for the most part are not readily aware of that and therefore they don’t present that way. Or they don’t share it that way”.

#### 3.2.2. Lilo i ka ʻenehana (Technology as a Distraction)

ʻŌpio described technology usage as both a means to escape and as a distraction. One ʻōpio claimed that for years all they “yearned” for was different kinds of technologies—phones, computers, gaming consoles, etc. These electronics provided their “escape,” which they, in turn, put “before everybody else”. Other ʻōpio echoed this constant use of technology, especially in relation to their cell phones.

At the same time ʻōpio, even those that expressed a desire for technology, viewed this same technology as a distraction and an uniquely modern day issue affecting their relationships. Most ʻōpio thought that technology “kind of took over”. Technology’s impact went beyond an individual’s escape and had farther reaching effects. Frustrated with technology’s hold, one ʻōpio stated, “It’s changing personalities. It’s changing how people do things. It’s taking them away from life. It’s a total distraction”. Another ʻōpio noted that unlike previous generations, technology distracts new generations from cultural practices, “It’s not the same, like, instead of going on your phone, [before] you can go to hula class”.

#### 3.2.3. Ka Hōʻalo (Avoidance)

Avoiding the thoughts of Native Hawaiian history or current issues came up in several ʻōpio talk stories. Some ʻōpio gave their history little to no thought, “I don’t really think of Hawaiian history”. Another ʻōpio didn’t know how to share thoughts or feelings about changes in Hawaiʻi, “I don’t really know how to feel. Because I don’t always think about that stuff”. Nonetheless, when pressed the ʻōpio knew of and were able to describe how Hawaiʻi has changed for NHs.

Some professional participants confirmed that in their experiences working with Hawaiian families many did not have a “super strong focus” on Hawaiian history or culture. Professionals also saw engagement of ʻōpio in school as a potential barrier to awareness of Hawaiian history. One ʻōpio shared that school curriculum was “not really anything Hawaiian”. A lawelawe believed that the schools “should be teaching history of Hawaiians or Hawaiian culture”. At the same time, another lawelawe countered by sharing that most of the ʻōpio they worked with did not know much about Hawaiian culture because they “didn’t really go to school”.

As if to increase avoidance, some ʻōpio would disassociate themselves from their history and culture saying that their Hawaiian ancestors, “didn’t really have, like, a lot of like clothes and like, like stuff that we have, you know like, [ ] an actual house and you know [ ] food”. Another ʻōpio shared that their uncle told them without Captain Cook, Hawaiians “Wouldda killed eachodda. All the [dead] under Kamehameha” implying that Hawaiians were violent in the past. Other ʻōpio agreed that Hawaiians were historically violent saying that in the past there was “a lot of fighting. [ ] People trying to take over, become chief”. Yet, another ʻōpio expressed disdain for how he perceived some Hawaiian to continue to behave violently saying “Everywhere I went … go to da store, somebody act Hawaiian”. Finally, one lawelawe shared, “everybody talk so highly of Hawaiians, but honestly we just like regula … like every other people. [V]iolence is in every one of us”.

#### 3.2.4. ʻImi Loaʻa (Consumerism)

Some ʻōpio expressed a strong desire for material items. One ʻōpio participant saw this increased consumerism beginning within the “new” generation, but being supported by older generations. In their eyes, the new generation were “spoiled brats” for always wanting or having the “new stuff” that they did not have growing up like phones, “nice slippers”, or “fake nails”. This drive for “new stuff” was also seen as a distraction from things once prioritized by Hawaiians like cultural practices or gathering together.

Another ʻōpio recognized the work needed to maintain a consumer lifestyle, “No matter what they say, ‘Oh yeah I get money, I get lifted truck, I get all dis, I get beach house…’ whatever. Yeah, and how you paying for ‘um? ‘Brah I get three jobs. Construction, welding, roofing.’ Big jobs, dangerous jobs they gotta take! Just to pay for all of these things”. Even with the amount of work put in, they did not see it as enough to keep up with the demands of consumerism and said that this cycle would eventually leave everyone “broke”.

### 3.3. Theme 3: He aliʻi ka ʻāina (Land Is a Chief)

ʻĀina was a central concept for many participants. The theme He aliʻi ka ʻāina is a well-known Hawaiian saying expressing a widely accepted Hawaiian belief of the importance of land compared to man. The participants discussed how Hawaiian land was stolen, desecrated, and disrespected by others, particularly tourists and settlers. Due to the continued abuse and misuse of ʻāina, participants identified the need for ʻāina to be returned to NHs. Four sub-themes were identified during these talk story sessions: Land Loss, Return Land, Land as Healing, and Environmental Destruction.

#### 3.3.1. Lilo ka ʻāina (Land Loss)

Loss of ʻāina was identified as one of the factors leading NHs to struggle. Participants shared that the ʻāina “stolen” and the resultant loss of collective land forced many NH to become houseless and live on the beach. One ʻōpio, when asked about her perspective of the loss of land, replied, “How we are doing is kinda struggling. Because my mom, we used to have a hard time. We were homeless at one point” This participant went on to share that “since [the settlers] took over, they make us pay like rent [ ] to live on our land … that we should’ve been living on for free”.

Lawelawe also discussed the impact that loss of land saying NHs “have suffered from these losses in a number of ways”. “A lot of Hawaiian families feel that they have been robbed of their land”. Another lawelawe shared how her kupuna had “the[ir] land [ ] taken away and now they’re suffering from this and now they’re marginalized”. Several participants tied the loss of land to food with one noting “where are you supposed to grow food?” if Hawaiians do not have access to fertile land. One ʻōpio shared that the loss of land impacted their ability to provide through hunting and fishing. “Like we cannot go hunting somewhere where we used to go hunting at without a white person saying that it’s private property. Or like we cannot go fishing somewhere without a white person telling us it’s private property”. Thus, participants connected loss of land to loss of cultural practices, loss of access to hunting and fishing areas, and loss of self-sufficiency. One lawelawe summed up this word well by saying “Land means a lot to Hawaiians. And so you, we can’t talk about historical trauma without talking about losing land”.

#### 3.3.2. Ka Hana ʻino i ka ʻāina (Environmental Destruction)

Historically, NHs live completely off of the ʻāina. Several ʻōpio noted that Hawaiians did not care about money, they cared about “the ʻāina”. One ʻōpio commented, “they used the land to um, to feed themselves. Like it was very self-sustaining”. This same participant went on to say, “most of the farms are like things that are not very sustainable. Like wheat and like soy and corn and pineapples. I’m pretty sure pineapples aren’t even native to Hawaiʻi. A lot of um, invasive things coming in”. The environmental destruction as well as use of foreign agricultural practices created an over-reliance on imported food and goods. One ʻōpio shared, “we have a lot of food shipped in”.

“Hawaiians did not care about money, they cared about “the ʻāina,” was a sentiment that several participants shared. Several ʻōpio expressed their belief that settlers valued things over money. ʻŌpio participants expressed anger and frustration by what they considered inconsiderate settlers and tourists. One participant shared “I see people trowing rubbish in the ground, I kinda feel pissed … I go tell them, pick up your trash, you know, pick up your mess”. Another ʻōpio said of tourists, they “come to our island and poison our water with sunscreen”. Yet, another ʻōpio shared that “no more fish cuz all the white people is over there trying to fish or their sunscreen is poisoning them”. One lawelawe summed up this value by saying “I am a true believer of malama ʻaina”.

#### 3.3.3. Hoʻihoʻi ea (Return Land)

In addition to loss of ʻāina, most participants discussed the importance of returning the ʻāina back to NHs. One ʻōpio simply shared, “I wish we could get Hawaiʻi back”. Another ʻōpio shared that I “really feel like one thing that can help our people is to [give] back their land”. Lawelawe shared the sentiment that returning land was appropriate with one saying, “the only thing we could do to make this right is give a whoooole bunch of land back”. Another lawelawe said, “We need our land back. You know, coz I really think that’s what gives people power is when they have land”. However, lawelawe also acknowledged the difficulties associated with unwinding colonization. In essence non-Hawaiians are hearing, “we’re [Hawaiians] gonna be sovereign, and you guys are gonna lose your land!”.

Despite the difficulties, ʻōpio expressed obstinance saying that it “doesn’t matter how many people can go against us and how many more areas of land they’re gonna take, they. We’re all still here. And one way or another, we’re gonna get our land back”. Lawelawe took this a step further discussing supporting the idea of “gain[ing] some sorta restitution or some sort of equity. Because the alternative is dey keep getting priced outta house and home”. Returning land was seen as the first step to leveling the socioeconomic inequalities that Hawaiians are currently struggling with.

#### 3.3.4. Ke Aloha ʻāina (Land as Healing)

In the most literal sense the ʻāina provides food, which participants identified as a major part of culture. One ʻōpio participant shared “food is a big part of the culture…. I always eating, like, dat kind of stuff [when] I grew up with um, my grandparents”. Another ʻōpio shared how the ʻāina fed his soul by sharing his “first time being in a loʻi patch. [ ] It was beautiful. [ ] It’s like really dirty though, like clogged up with mud and stuff. So we went over there, and we cleaned up um, California grass. And we was in the mud, and I was lovin’ it! I was lovin’ it cuz I would be in the sun. I love the sun, and I love the food”.

Several participants discussed how ʻāina can be used as a pathway to healing, especially when the entire ʻohana is involved. For example, one participant recounted how during NH culturally based afterschool program “parents are invited to come to the āina and work and eat from it. It’s not just the keiki”. They went on to say that ʻāina-based programs have been tremendously successful [i]n helping the children see themselves differently [and in] the way they behave and engage with one another”. One lawelawe discussed another program where “they utilize restorative circles, they work the ʻāina every week. They make sure [the participants are] in the soil … in the lepo (dirt). And that’s been extremely healing”.

### 3.4. Theme 4: Kū i ka Welo (Family Connectedness)

Kū i ka welo emerged as a major theme and refers to vertical and horizontal connectedness. To be kū is to stand or reflect a family trait while welo is a family tradition. Thus, this theme articulates that deeply seated connection that participants had to their immediate family, extended family, and community. This fourth theme included four subthemes: Family support system, Distortion of parental values, Connectedness to extended ʻohana and communities, and Kupuna.

#### 3.4.1. Ka ʻohana (Family Support System)

Whether or not participants had a connection to their ka ʻohana played a hypercritical position in their lives. ʻŌpio reminisced about how “we could call all our ʻohana come get together” and barbeque. Many of the ʻōpio indicated that their parents and older siblings would tell them stories and show them how things should be done. Several participants discussed living in multigenerational homes. In part this is due to the high cost of housing, but as one lawelawe shared these multigenerational homes hold “a lot of support and love! And [ ] children are learning from the generations before. That is the gift”.

However, several participants mentioned the breakdown of the ʻohana. One ʻōpio commented that “nowadays, I see the Hawaiians, they all separated. So like, they barely see their own, their ʻohana”. This was reiterated by a lawelawe who shared that many of their ʻōpio clients “either don’t have a family or they’re just disconnected from them. [ ] You know, they’re running away, living on the street”. Many of the ʻōpio had stories of running away whether briefly or for longer periods in an attempt to remove themselves from stressful family situations.

When speaking about their ʻohana, one ʻōpio declared, “Family is big for Hawaiians,” not only because they care, but they “would protect us”. Other ʻōpio reiterated feelings of support and protection by their immediate ka ʻohana and ancestors both physically and spiritually, “I feel like I’m moving and protected. No matter where I go. I’m being watched like in a good way”. Unfortunately, as important as family is sometimes family did not live up to expectations. One lawelawe shared that when they had to take ʻōpio into protective custody, the ʻōpio “were very disappointed in their caregiver. Whether it be parent, grandparent, aunty, whoever was caring for them at the time”. They went on to say that this was especially prominent when their caregiver “was not supportive of them” or ʻdid not believe them”. Another lawelawe shared that “family is always invited … [but] it’s so rare we even get the families to even come out even try to participate”.

Lawelawe also elaborated on the magnitude of the ʻohana and it’s role in other aspects of NH life, “Family connections, well-being, spiritual well-being, cultural identity, those are things that I see as helpful and helping to address the negative impacts of historical trauma”. Lawelawe expressed a desire for more family-oriented programs rather than just focusing on the ʻōpio with one saying, “we need more programs specifically that focuses on the family component”. Another shared that they appreciated one program where they would, “go[] to the loʻi and educat[e] the families about all that. It also, um, shows the kids how their parents will put in work”. “ʻOhana is crucial!” was how one lawelawe indicated that ʻohana was a necessary ingredient to healing trauma.

#### 3.4.2. Ka Hilihewa o ka Makua i ke ʻano Haole (Distortion of Parental Values)

The application of western standards of child rearing has effectively weaponized and disrupted traditional parental values. Several ʻōpio discussed the importance of “honoring your parents”. Nā lawelawe, however, framed this value as “respect” noting that “everybody would have respect for one another” and that “respect was earned”. Several ʻōpio also discussed “lickins” or spanking often with a “rubbah slippah” as a traditional Hawaiian parenting style. One ʻōpio said “lickins [is] not just punishment, but supposed to make you a better person”. One non-Hawaiian lawelawe shared the belief that some Hawaiian “parents are like old school style, so they don’t you know, they show affection but sometimes you know, this hard knock style”. However, a Hawaiian lawelawe shared that “lickins was a way that they discipline, they discipline their children,” but it was taught by the first missionaries who were “strict and most extreme”. They went on to say that Hawaiians “think this is a practice that our people had from 500 years ago,” but it was not.

One ʻōpio acknowledged that parenting styles in the past were different. She shared that “the way parenting is now was way different than back, before”. Another ʻōpio shared that their caregivers would share stories about ghosts or menehune that would harm children that did not behave, so that they were scared to misbehave. They went on to share how they have more freedom than their parents’ generation because their parents are not often at home.

#### 3.4.3. Ke Kauhale (Connectedness to Extended ʻOhana and Community)

There was a consensus among all participants on the importance of being connected to extended ʻohana and community. One ʻōpio shared that she was adopted “at 8 months old to a very Hawaiian family”. She defined family as “where you take care of one another” and shared that they “taught me everything I know about Hawaiian [] culture”. ʻŌpio shared that in their community people were not “selfish” and that “we’re all trying to help each other out”. Another said that in their community “everybody knew each other” and “we don’t wanna see each other fall”.

One ʻōpio discussed how she found community in her hula halau (school). Another ʻōpio shared how his aunty and uncles passed down a lot of cultural knowledge to them. For example, “my aunty she taught me like make lei” and “my other aunty’s is a singer. I have a bunch of them that play the ukulele and my uncle wrote me a song”. Uncles also played a large role in storytelling with many ʻōpio sharing that their uncles would tell them scary stories to encourage appropriate behavior.

Lawelawe statements aligned with the ʻōpio and additionally described the strength of having extended family connections. One lawelawe shared her belief that “the thing that separates me from my clients who have nowhere to go is in my community the people that I know, the people that I trust, they don’t have that”. Another lawelawe said that we need to “help our ʻōpio hold on to those connections”. One lawelawe suggested that there be “more halaus” while another suggested that the focus needed to be on “building community” so that ʻōpio would have more support systems to draw upon, especially during crisis.

#### 3.4.4. Ke Kupuna (Elders)

The last sub-theme identified was ke kupuna translated as an elder, grandparent, or ancestor. One ʻōpio reminisced about her childhood defining who her ke kupuna is, “my ancestors is my ʻaumakuas (personal/family gods) but my ʻaumakua is like my kupunas like my adults, like my my mom, my aunties, my uncles”. Kupuna were often discussed in terms of bringing the family together or seen as “the matriarch” holding the family together.

Many participants during the talk story sessions highlighted ke kupuna and the crucial role in their lives. In many instances ʻōpio described being raised in part or completely by their kupuna. One participant shared that their kupuna had “Knowledge from their upbringings … [that] they like to try and just pass it through generations”. One ʻōpio shared that their kupuna “did all that hard work” teaching their parents and them cultural practices. They went on to say, “I still remember those days like [ ] laying net and [dancing] hula”.

However, stories from their kupuna’s generation sometime negatively impacted the ʻōpio. One participant shared how they “would talk about the shame of being Hawaiian”. Another participant said “because my older family is still alive and they still talk about like my great grandparents and how [changes from colonization] affected them as children”. Another lawelawe went further and said that sometimes people “tell[] us, that was in the past. You know, get over it! … [but] I would say, you know what? Happened in my grandfather’s day. That wasn’t much of the past”.

### 3.5. Theme 5: Hoʻohikihiki (Dreams, but Not for Me)

Hoʻohikihiki translated as to go through the rigors of reaching a goal with no promise of getting it represents the sentiment that both ʻōpio and lawelawe shared. Participants discussed recognizing that Hawaiians thrived in the past, but that many of the ideals that people strive for today are not meant for them. Money, housing, and intellectualism were all seen as things that were not for Hawaiians. At the same time participants were proud of the abundance and communal living that existed prior to colonization. This divergent thought process suggests that a shift occurred in how these participants saw NH in society.

#### 3.5.1. Ka Nohona (Thriving in the Past)

Participants agreed that Hawaiian ancestors thrived in the past. Many understood that their ancestors had sufficient resources to not only sustain themselves, but thrive. Several ʻōpio participants noted that their ancestors were able to sustain themselves by hunting, fishing, and growing kalo; however, these practices had since diminished. One participant shared, “If white people didn’t come here, I think everyone would be healthy. Healthy and happy because we can go hunting, fishing, anywhere we want. Because all the spots we can go now which is not private property, is no more fish cuz all the white people is over there trying to fish or their sunscreen is poisoning them”.

Several other participants echoed the belief that Hawaiians were healthier in the past. One participant shared that Hawaiians did not care about money, they cared about “being healthy and making sure everyone stayed healthy”. Another personalized it saying, “I feel like we would be more healthy cuz … I like to be on my phone all the time and just eat. But if it was back then, I woulda been healthy”.

While a few participants idealized the past, the majority noted that their Hawaiian ancestors had to work hard. They noted that Hawaiian leaders of the past were “strong,” that they “respect[ed] other people,” and that they “was doing more”. One participant contrasted the past with what they saw today as “chaos,” but in the past “everyone would be doing their ‘ting. Like clearing, doing more taro patches. Fish. Gathering fish”. The orderly division of labor was brought up by another participant who said, “things would be more organized. There would be more consistency” if Hawaiian leaders from the past were in charge. Finally, participants shared the belief that “before white people came, there was no racism” and that it was a time of “peace”. Finally, one ʻōpio connected western contact to losing self-sufficiency by saying “We didn’t need them to help us with the fishing, hunting, but then when Captain whatever-his-name-was came to our island, everything just went down from there”.

#### 3.5.2. Ka Waiwai Kālā (Money/Haves and Have Nots)

When asked if there were things they had to deal with that their ancestors did not, nearly all participants said, “The biggest thing is money”. ʻŌpio participants shared things like, “I cannot pay rent, none of us [family members] can pay rent”. At the same time ʻōpio participants did not desire a return to pre-contact times. While there was an appreciation of the order that Hawaiians had as a society, they recognized the comforts associated with modern living. ʻŌpio participants, especially, shared that moving back to a system from the past would require a significant increase in work.

Nonetheless, there were elements of Hawaiian history that nearly all participants discussed as positive. “A lot of Hawaiian culture is to take care of other people too, so I think if [ ] we still had a monarchy, more people would be taken care of”. Several participants discussed how Hawaiian ʻohana and communities were close-knit before contact. They shared the understanding that Hawaiians would barter with each other sharing resources so that everyone would have what they needed and no one had to go without. One ʻōpio noted that “the food problem and the money problem kind of go together”. While another discussed access to food saying that today food did not require as much manual labor, but Hawaiians “trad[ed] the, you know, easy food for peace”. When one participant was asked what was different about modern times they simply said, “poverty,” leading another to say “poverty wouldn’t be a thing”. Thus, participants overwhelmingly shared the opinion that poverty was new.

Participants articulated the belief that modern day politicians do not care about Hawaiians evidenced by the lack of attention to Hawaiian issues. In addition to politicians, ʻōpio, especially, believed that they were being “disrespected” by authority figures. One ʻōpio shared that “people talk down on us”. Lawelawe participants also discussed trying to mediate between authority figures and NH ʻōpio, especially around the issue of “respect”.

Several participants also contrasted their situation to those of tourists and other Hawaiʻi residents stating, “a lot of families, a lot of my friends that I talk to, they have to have multiple forms of income just to afford living here[.] Whereas I’ve seen some people who have it made in Hawaii Kai and all they do is, like, realty. And they barely have to lift a finger, and they make enough to have a three-bedroom house”. Thus, participants made a distinction between their situation and others often, though not always, suggesting that the difference was unfair.

Tourism frequency came up when discussing money problems, poverty, and differentiated situations. Lawelawe validated that tourism was also a frequent topic in their work with NH ʻōpio. Many ʻōpio blamed tourists for their situation saying that tourists “takes up a lot of space” and “they bring, like, diseases”. Another participant noted that “people are not so friendly to tourists, especially some of my Native Hawaiian friends. They really don’t like tourists”. Thus, participants made a connection between tourism and NH socioeconomic status.

#### 3.5.3. Nā Hale (Housing)

Hawaiʻi’s high cost of living was seen as particularly troublesome. Nearly all participants linked the high cost of housing to individual and community struggles. Many of the ʻōpio were housing insecure with some living in residential housing programs and others having experienced living in cars or “on the beaches”. “A lot of people cannot afford to live here anymore because it’s getting very expensive,” was a common sentiment heard among participants.

Many participants, including lawelawe, felt that they would never be able to afford to purchase a home of their own. One participant summed it up by saying, “it’s so expensive, nobody can afford even buy one house here”. Several ʻōpio talked about working even though they were still minors. One ʻōpio said that they “work so my mom can forget about money. [So] she doesn’t focus on money”. Lawelawe shared stories of ʻōpio they worked with working legal and illegal jobs in order to help support their families sometimes at the expense of schooling.

Several participants shared that the high cost of housing in Hawaiʻi has led to extended family and friends moving away. “I kind of resented [ ] the news [ ] calling uh, Vegas the 9th Island. And at first I thought it was kind of funny, but I’m thinking Hawaiians are being forced to move there”. One participant discussed Hawaiians that moved to the mainland as “kinda sad that they’re just going to be just vacationing here”. The displacement of Hawaiians to the mainland for cheaper housing and better job opportunities was discussed with sadness and was associated with losing ties and connection. However, at least one ʻōpio thought that Hawaiians that moved away were “fucking crazy” because “from [her] experience being like, you know, on the island, to going to the mainland … was really traumatizing … to not see salt water”. Bringing out the connection that many of the ʻōpio have to the ʻāina. At the same time, however, participants acknowledged that individuals had to “do what they gotta do”.

#### 3.5.4. Ka Hōʻole Kahuna (Anti-Intellectualism)

Many lawelawe shared their resonant belief that “Hawaiians were intellectuals”. Yet, they acknowledged that they grew up having to hide their curiosity or downplay their interest in academics. Family and friends would “mock” their “inquisitiveness” or accuse them of “thinking they were better than them”. One insightful participant said, “it also plays right into the hands of the existing power structure for us not to be curious”. He went on to say, “And so you have to wonder who set that narrative. We … they may not be the ones saying it out loud now, we may be saying it to ourself. But who set that culture of doing that? I doubt it was our aliʻi or our kupuna”.

Lawelawe also shared stories of some of the NH ʻōpio that they worked with trying to share and implement what they learned only to have their family dismiss them. In one case, the child’s father threw him out because the ʻōpio was trying to change the “status quo”. Another lawelawe had to promise one of her ʻōpio that she would not tell others that they enjoyed reading. This participant said of the ʻōpio “everybody starts turning down their lights … in front of all their friends or in front of their family, they turn it all down … and then the times when they don’t turn it down [ ] someone in their family will make fun of them”. NHs who do exhibit their intellect are often confronted with, “Who you think you?” The participant went on to say that in other households “when a kid knows a fancy word, they [the family] go wow, impressive. They don’t go, who you ‘tink you? … that’s not who our ancestors were”.

Anti-intellectualism also bleeds into the educational system in Hawaiʻi, which appeared prominently across the talk stories. In Theme 6, we discuss the state of the educational system, but want to highlight here that many participants felt that the educational system did not provide appropriate education. One lawelawe shared that she “d[i]n’t think a lot of the kids that [she] worked with actually knew a lot about Hawaiian history”. Another participant who worked in the school system would hear that NH kids “talked too much in class” or “neva do their homework,” yet the school would continue to pass the student so that you would have a “9th grader [whose] last grade that they actually finished was 5th or maybe 6th”.

The idea that some things that are not meant for Hawaiians seeps into other areas such as homeownership. One participant shared that even though they were doing well they still did not think purchasing a home was something in their future. This aligns with another participant who likened it to a type of “cognitive dissonance” among NHs that certain things were just not in the cards for them. The participant shared that after reflecting on his situation he recognized that his resistance to purchasing a home was “historical trauma. [ ] It’s so fixed in my head like I’ll never own. Right? [ ] That’s not what we do. We don’t. We fix the houses. We build the houses. We don’t fucking buy the houses”.

### 3.6. Theme 6: ʻŌnaehana Huikaulua (Messy System)

The theme of ʻŌnaehana huikaulua or messy/confused system refers to the colloquial “powerful government of social organization that controls people’s lives”. There are seven sub-themes, each of which represents a smaller system, that NH participants imbued with some type of control over their lives. Colonialism, which brought tourism and militarism, was seen as central to the ʻŌnaehana huikaulua. The role that schools played and continue to play in perpetuating colonization and re-traumatizing ʻōpio was brought up by many participants. Other institutions, such as Child Welfare Service, the healthcare system, and the carceral system, were provided as additional evidence that the current system does not have the best intentions for NHs.

#### 3.6.1. Ka Pono Waiwai Haole (Colonialism and Other-Isms)

The majority of participants discussed the illegal overthrow of the Hawaiian Kingdom and traced many NH woes to that time. Colonization was discussed not just in the past tense, but also the present tense. All of the ʻōpio articulated an understanding that NHs lost control over their lands through colonization, however, only approximately half of the ʻōpio articulated the view that those currently in power inherited or benefitted from the overthrow of the Hawaiian Kingdom. ʻŌpio that engaged in cultural practices such as oli (chanting), hula, or mele (singing) or attended a Kaiapuni school (Hawaiian immersion school) for part of their education were more likely to connect colonization to present day inequities. One participant summed it up as, “I feel like you will be suffering in today’s society just knowing that life for our Native Hawaiians could have been very different if we weren’t overthrown”.

While many participants identified colonialism and other -isms as an issue, they also acknowledged that “colonization and the powers that be, [they] don’t want to change”. Participants discussed that because they were brought up in this system, there was a fear of “disrupting the status quo” and one participant even said that this fear stemmed from a concern that “things will get way worse than they already are” if they spoke up. Thus, a type of self-colonization was also acknowledged by some. One participant said, being economically successful creates an “extreme fear of the systems, right? Whether it’s the banking system, whether it’s healthcare, [or] it’s [ ] extreme distrust”.

With colonization came the missionary system, which played a heavy hand in assimilating NHs. Nearly all the ʻōpio identified religion as a mechanism that changed their ancestors’ way of life. Hawaiian culture was seen as “more open, you know, and even back den, māhūs [homosexual/hermaphrodite] were accepted”. In contrast, participants saw Christianity as “strict” and “oppressive”.

Another system of institutional oppression came from banning ʻōlelo Hawaiʻi (Hawaiian language). One ʻōpio shared that “My grandma said, [ ] she was [ ] talking in Hawaiian to her dad and the teacher said not to do it. And told the dad not to do it anymore. And ever since then, she said all they spoke was English”. When asked how they felt about this the ʻōpio responded, “I can’t explain it … It’s empty. It’s empty, like … there’s nothing to feel about it”. Nevertheless, many participants did not share feelings of being part of the larger society, but rather felt that they were “othered,” “receive bad service,” and were subjected to “racism”.

#### 3.6.2. Ka Malihini (Tourism)

The impact of tourism came up in many of the talk stories. ʻŌpio expressed animosity towards tourists who “take[] up a lot space”. and are “rude … refusing to wear masks and, you know, not caring”. One ʻōpio said they felt “like, we were kinda robbed, and it’s the tourist fault”. He went on to say that he believed that some crimes were targeted at tourists because “it’s related to, like, wealth … not having wealth, [and them] having the nice parts of our island”. Another participant said “the tourist [is] all over my stuff”. One lawelawe shared that friends visiting from the mainland felt like “they [are being] stared down [when at the beach]. They can feel the tension. Guys just want to fight them”. This participant went on to say, “so, we don’t always have a lot of aloha for tourists”. Another participant suggested that it was important to educate tourists about the “language, [and] make sure that [ ] people know [the] history, practices”.

One participant linked visitor accommodations and vacation rentals to the high cost of housing. Another lawelawe suggested that a “moratorium on visitor accommodations” was needed. This participant was referring to an effort to prevent off-islander investors from purchasing houses and condominiums in Hawaiʻi in order to rent them as short-term rentals to tourists, and thus, removing them from the resident housing market [50]. They noted that even residents making decent money have a difficult time purchasing a home because “people from the mainland [ ] are buying property specifically to use it as vacation rentals. They’re using it to make a profit on Hawaiʻi”. Thus, tourists were perceived as part of a system that was taking over Hawaiian land, making rental prices skyrocket, and forcing more Hawaiians to give up their places of respite.

#### 3.6.3. Ka Pūali Koa (Militarism)

Participants often brought up the connection between the military and their feelings of historical loss. The military has had a significant presence in Hawaiʻi for many years. One lawelawe shared that in the past, many Hawaiian high school graduates were funneled into the military saying, “you either get a full-time job or you going to the military”. Another participant discussed how much land the military controls in Hawaiʻi. They noted that “people want to get rid of the military bases” as a way to return land to Hawaiians.

Finally, several participants mentioned Red Hill, which is a Navy fuel tank installation that was found to be leaking fuel into the community’s water supply and is the subject of numerous recent protests. Participants provided Red Hill as an example of the government not caring for Hawaiians or the ʻāina. One participant said, “Red Hill… Not long time ago, but back then they did it before and it’s happening again”.

#### 3.6.4. Ka ʻehaʻeha ma ke Kula (School Re-Traumatizes)

The impact of schools cannot be overstated. Participants discussed how schooling in the past inflicted trauma and how schools today are passively perpetuating historical trauma and sometimes actively re-traumatizing NH ʻōpio. Many participants shared stories of frustrated teachers who lost control and said inappropriate things to NH ʻōpio. One participant shared that NH ʻōpio were told “they would never do anything but like, work in the fields or like do landscaping on the hotels”. Another social worker shared an incident where a teacher yelled at a NH boy “in fron of the class, saying he was stupid and dumb [and] heʻll never graduate from high school”.

Some lawelawe recall NH students telling them “how miserable they are in school and how poorly they’re treated”. Once they were in a safe environment these ʻōpio would share “how badly they feel about themselves and how hopeless they feel”. One ʻōpio who thrived in another environment “flash[ed] back to something [their] second grade teacher told [them] when [taking a] test”. Another participant shared another traumatic story of when “one teacher made [a] little Hawaiian girl sit like a dog on the classroom floor cuz that’s what she was. That’s how she behaved, like an animal. And like everyone in the school knew it. I mean, the class heard it”.

While some teachers appear to lack empathy, even teachers who care deeply about NH students are confronted with students that bring anger from prior mistreatment into their interactions. One such participant shared that “some of the boys have been saying they [other teachers] don’t care because it’s just another kid from Waimanalo”. These were NH middle school students who were referring to a missing person case that received significant media attention. Amber Kalua, a NH girl, who’s foster parents were found to have caused her death by keeping her in a dog cage in the bathroom. While this story was widely covered by the media some teachers, counselors, and administrators had difficulty handling the issue with students.

Many of the participants reflected on the impact of the lack of local teachers and policy of recruiting teachers from the mainland. And they’re not comfortable. They’re all young, and they don’t understand their kids”. School counselors and psychiatrists are “trying to build [ ] capacity for our teachers to feel comfortable at least addressing certain issues”. However, they believed that historical trauma needs to be built into the teaching curriculum and part of system-wide training leading one participant to share that they “really wish[ed] we had a more culturally-sensitive superintendent”.

Another issue was that some long-time teachers would conflate a NH student with their parents, siblings, or other relatives, especially if their family member had “caused trouble” in their eyes. One school counselor shared that, “when we be talking about incoming freshmen or even at the elementary level, they’d be like, oh, so and so, oh that’s one pilau [rotten or to stink] family. Goin’ be just like the older brother. So they’re already labeled…. They already came into the system lost”. Another participant shared a similar story with a teacher saying, “Oh I remember your uncle!” From that point on the student was unable to break away from his uncle’s reputation. The students “get traumatize and retraumatize many times because of something that happened to someone else, and especially when it comes if you have a parent or a sibling or a family member who’s been incarcerated. And they’ll be like oh you goin’ end up just like your fatha. You goin’ end up just like your uncle or your mother”. The result are NH ʻōpio who are “already walking around school saying, just put me in prison already”.

The “classic school-to-prison pipeline” was brought up by several lawelawe. They noted that while things have gotten better in some schools, in other schools they recalled “HPD [Honolulu Police Department] being there daily”. The punitive nature of schools was identified as an area that needed to change. As one participant put it “school systems are not juvenile [detention] systems … the system is not working”. One participant speaking to the differential treatment of NH and other ʻōpio, said that schools were “designed [ ] for this outcome”. Another lawelawe shared that “every time I convince a parent to send their kid back to school. Every time I convince a child to go back to school when they’re getting hurt at school, like … Ugh”. This lawelawe was expressing the difficult position they were in knowing that the ʻōpio had to attend school, but also understanding they now played a role in perpetuating trauma.

Parents, on the other hand, “just [do] not feel[] comfortable coming to the school and talking to teachers…. They know their kids [don’t like] school. They make their kids go to school, but then because they didn’t have an easy time with the education that was put on them [ ] they just don’t feel comfortable, and so they just kind of avoid dealing with any of the school things”. Another lawelawe shared a more personal comment by saying that they had to move their children to a different school because the “school makes my kids feel like I did … like trash”. Even after an entire generation, many NH ʻōpio are still facing significant bias in an institution that is supposed to promote their success.

#### 3.6.5. Nā Kōkua Kamaliʻi (Child Welfare Services)

Many ʻōpio and lawelawe brought up the Child Welfare System and its impact on their lives and on their community. One lawelawe shared “Everybody has this negative um, negative what do you call? Negative thought about child welfare”. This was in part because “Native Hawaiians are overrepresented in the Child Welfare System,” but in part because many Hawaiians felt that “I just don’t tink we should be taken away from our families. It’s not the culture”. A lawelawe shared that because of the deep connection to ʻohana some “kids get abused and all of that kind of stuff. But they don’t report it. They don’t wanna report it right cuz they’re scared of getting taken away”. Another lawelawe said, ““the fact that you separate a child and parent so they can heal is not something our kupuna would do. You know, it’s not intuitive, it’s not natural, it goes against generations and generations of our DNA”.

Some lawelawe acknowledged that there may be instances where it is necessary to remove a child from their home, but ʻōpio expressed a desire to be placed with another family member instead. One lawelawe shared that as “an investigator, I pretty much as in and out of a case” leading one ʻōpio to question whether CWS cared about them. Another ʻōpio who held significant anger shared that CWS removed them, but she felt they did not do their job because she should have been placed in a “safer foster home[]”. One lawelawe spoke about how many NH ʻōpio were involved in multiple government institutions like incarceration and foster care. She shared that she was involved in a transitional housing program from NH ʻōpio age 18–26 that ultimately housed 45 NHs and “all but one had been in foster placement and at one point”.

Many participants expressed how this system had taken them away from their families creating another compounding loss and how this has created an intergenerational cycle of trauma from an institution whose intention was supposed to protect ʻōpio. One participant discussed NH ʻōpio identity, feelings of worthiness, disconnection from self, family, and community saying, “they act out in different ways. [ ] Whether it’s suicidal ideation or entering into domestic violence relationships, drug usage, overreliance on alcohol, or sexual promiscuity, or going from job to job or relationship to relationship, not being able to complete school. So there are a number of things that occur because of that deep grief or trauma or kaumaha”. She went on to connect this to the impacts of the illegal overthrow of the Hawaiian Kingdom saying that “being separated from family and being from families that have that deep kaumaha themselves” only compounds.

#### 3.6.6. Ke Olakino Kauʻewa (Health Disparities/COVID)

Health inequities were seen as something new. Our ancestors “thrived until non-Hawaiians came. And then they brought us disease and uh slavery”. One NH ʻōpio shared that their mother was unable to work because of “medical issues”. That same ʻōpio’s father was already “on his canoe ride,” signaling that he had passed. ʻŌpio saw family members struggle financially because “we cannot do anything without the body functioning. It’s hard”. Yet, at the same time, not all NHs have access to health care, especially if they are not employed.

Several participants linked the current COVID-19 pandemic to prior waves of disease brought during early contact. One lawelawe said, “the missionaries knew what was happening, but they turned it on the Hawaiians and say well, it’s because you guys do not live good lives. So that is why you guys are dying. [T]hat is exactly what is happening with COVID today in Hawaiʻi. [T]hey’re allowing people to come here, fly here, unsafely and yet they’re blaming the residents”. One ʻōpio shared that COVID took several of their ohana, saying “day by day one died, next day, other one died”.

Another participant shared that when epidemics occurred in the past, Queen Liliʻuokalani would implement a “shut down. All the visitors, nobody can come in and out”. They felt that Hawaiians were “dying from COVID, and their lives could have been saved if they had just shut down”. Yet another participant recalled that during a smallpox epidemic the aliʻi would require 40 days of “quarantine before stepping onto our lands”. COVID is seen as another coming of the same epidemics that killed so many ancestors and quite a few participants felt that Hawaiians would have fared better if the state had taken an approach that mirrored what the Queen and aliʻi had done in the past. As one participant shared, “Like Christians will say, well what would Jesus have done? Well what would our aliʻi have done? And so you look at what would the aliʻi have done? They would have protected the Hawaiians”.

#### 3.6.7. Ka Hoʻopaʻa Hao (Carceral System)

Most of the NH ʻōpio participants were involved with the juvenile justice system and nearly a third were housed in juvenile detention centers. Many shared statements like if “Hawaiians still governed, [ ] they probably wouldn’t have lock down facilities like this,” “this [facility] shouldn’t even be a place,” and “this wouldn’t be somewhere where anybody would have to go,” indicating their deep rooted belief that their situation did not align with the values of Hawaiians.

Lawelawe shared that “for Native Hawaiian youth, we know the stats. We know they’re more likely to be incarcerated, we know they’re more likely to be arrested, we know that when they do get incarcerated they’re more likely to serve a longer sentence and less likely to get out on appeal”. Others reiterated that “we have to reform the criminal justice system”. Another lawelawe linked incarceration to poverty and violence, sharing the story of a NH ʻōpio whose mother was the victim of domestic violence and the ʻōpio started engaging in illegal activities to help pay for rent so they could move out on their own. Thus, reforming the criminal justice system may not be sufficient without also reforming numerous overlapping social structures and support systems. Because, as one participant said, “every system is set up to be disadvantageous to Native Hawaiians. Period. Flat out”.

### 3.7. Theme 7: Hihiha Hīkākā (Entangled, Sprawling Trouble)

Another powerful sub-theme that materialized during various talk story sessions with participants was Hihia hīkākā or entangled sprawling trouble. Nā pilikia kaialu or community conflict spoke to members of the NH community fighting with one another over political views, which participants attributed to historical trauma. In addition, many participants discussed nā pilikia kūloko or internal conflict within an individual.

#### 3.7.1. Nā Pilikia Kaialu (Community Conflict)

Several participants commented on the conflict within the NH community and how community was in disarray. Several ʻōpio expressed frustration at how the NH community could not agree on certain things. One ʻōpio who was especially frustrated with NH adults who focused on protest said “how people can just drop what they’re doing and go and do stuff like the Mauna, the turbines, the railway”. Each of these were large scale protests spearheaded by a certain segment of the NH community. The ʻōpio went on to say but “how much people voted for the railway” suggesting that these NHs may not have voted or that there were other NHs who supported these issues. Another participant agreed saying “We don’t have a whole lot of Hawaiian. Period. And if Hawaiians don’t agree, you have diluted that little bit of people. And Hawaiians aren’t agreeing. So that’s a huge barrier”.

One participant stated that the, “community would be a lot more unified,” if they recognized it’s loss or disconnection from the NH culture due to being too “Americanized”. This participant vocalized how individualistic NHs have become and how far removed they are from NH collectivistic practices. This participant shared that, “I think it kind of ruined our culture in a way where people are just fending for themselves and you know, [ ] everyone’s divided and kind of [experiencing] chaos in a way”.

Additionally, lawelawe discussed how NH’s have also chosen to change their appearance. One lawelawe stated changing, “our hair, our clothes, our weight” to look like “whiteness” and if not “we aren’t good enough” and “that is infused in everything”. This lawelawe noted how these changes can be, “subtle” or “sometimes it’s obvious,” but it says the same thing, “you’re not good enough”. Conversely, some NHs shame other NHs for not being Hawaiian enough. For example, one lawelawe stated, if a NH does not, “say a Hawaiian words properly, or you don’t [wear] your lei the right way, or you mistakenly don’t make the food the right way,” then you’re not “Hawaiian enough”.

#### 3.7.2. Nā Pilikia Kūloko (Internal Conflict)

Most of the lawelawe witnessed internal conflicts within the ʻōpio ranging from loss of cultural identity, blood quantum, and lack of self-worth. Many NH ʻōpio and lawelawe articulated a desire to learn about their culture. While all expressed pride in being NH, many did not feel confident in their cultural knowledge. This is not surprising because as one lawelawe shared, “from being decimated population-wise to being relegated to forced assimilation against losing their nationality and country, identity affiliation with culture … sovereignty over land, loss of language” it is no wonder that NHs today are not as culturally knowledgeable as in the past.

Another lawelawe mentioned, “asking, not [only] the questions about historical trauma, but about [NH] identity,” is critical when working with ʻōpio. In many cases, “they [NH youth] don’t make the connection[s] or connect the dots”. NH ʻōpio “are not readily aware of that and therefore they don’t present that way. Or they don’t share it that way”. Another lawelawe observed NH ʻōpio’s behavioral issues due to a, “lack of identity, and so [ ] they’re constantly searching for an identity”. One ʻōpio shared that “you gon’ be confused everywhere you go. No matter who you talk to”. referring to hearing differing versions of NH history.

Several NH ʻōpio introduced the topic of blood quantum or “bloodline” along with the fear that the NH population and culture was becoming extinct. One ʻōpio encapsulated this by saying the NH, “bloodline is deteriorating” and “now it’s just like going down, the percentage and their blood”. Another declared, “I notice that younger people who identify as NH, their blood quantum is always less than 50%. There’s not many 100% full NHs anymore” and “I feel like being NH, your blood quantum goes down and down as generations go on, so there might not even be Hawaiian culture, who knows, right? It might die out”. At the same time ʻōpio expressed a desire to learn more with one ʻōpio saying “Cuz I basically diluted as hell, and like I’m like Korean, Vietnamese, Asian, and like all that stuff and Hawaiian. So I don’t act Hawaiian. I wish I knew more. But I just don’t”.

Finally, NH self-worth was an apparent issue experienced by both ʻōpio and lawelawe. One ʻōpio commented that NHs “don’t know their worth. They don’t know what they can become or who they are”. Similarly, a lawelawe suggested that some NH ʻōpio base their worth on having children and a family, “If they can just have a baby with someone,” it can, “save them from their family and make them worth something,” and make them feel “worthy”. This lawelawe acknowledged how this is true for women in other cultures as well, but, she does not know, “how it got twisted where, um, every Hawaiian girl I met believes, like, that she has to have a baby and that that’s gonna make her life and make her valuable”.

### 3.8. Theme 8: Hoʻi i ka Pono (Return to Balance)

The final theme of Hoʻi i ka pono articulates a strengths-focused narrative and was inspired by a participant who stated that we needed to “changed hearts and minds” in order to tackle Native Hawaiian historical trauma. Hoʻi i ka pono literally means to return to a state of balance, which aligns with participants’ discussion of re-empowering, revitalizing, and re-centering on Hawaiian values by re-centering education around culturally responsive programs, re-empowering the community to unify in efforts to advocate and engage, re-engaging and growing leaders and revitalizing a self-sufficient future.

#### 3.8.1. Ka Hoʻomana (Empower Education)

In contrast to the current educational system, both ʻōpio and lawelawe envisioned an educational system that uplifted NH ʻōpio. “We don’t need another curriculum from the mainland”. Instead, participants articulated a need for a culturally responsive curriculum that was not only trauma-informed, but also incorporated “healing historical trauma”. Incorporating Native Hawaiian history was seen as a minimum, but having all students “learn the kumulipo” and ʻōlelo Hawaiʻi would elevate Hawaiian ways of knowing. One participant said that “the one thing that could unite us, the one thing that could unite all people of all descents, of all races, of all countries is ʻōlelo Hawaiʻi and the culture”. However, “people still seem to be resistant to that”.

Lawelawe envisioned an educational system that incorporated “ʻāina-based learning” with students spending time in the loʻi and planting. One participant went so far as to say that “every school should have a garden. That’s where healing takes place!” Practitioners also discussed reintegrating the practice of hoʻoponopono, a traditional method of dispute resolution. This led one participant to argue for bringing hoʻopono, which does “not to go so deep into the trauma like how hoʻoponopono does, but [still uses] a cultural lens [to] help families to resolve, um, conflict within the home” to the schools. Participants noted that prior to contact, Hawaiians lived “pono” or in balance, but conflict creates imbalance, which needs to be restored to promote healing.

Other participants wanted to see “more halau. [ ] We need to elevate the culture”. Another participant shared some of the work that had been done with the women’s correctional facility where hula was used for healing. Hula allows you to “breath in … inhale who you are as a person. If your kumu is good, before you even take a step, you need to understand, you need to center yourself”. Thus, bringing culture to the educational system was seen as a way to empower healing in schools.

One participant summed up this theme by saying, “it’s recognizing that duality [between strength of our people and what it’s taken for us to survive] because what we had was great, who we were was great, and the reason why people are so afraid of us is because of our power. Which is a very different way to sit than I used to think, like, poor us. We’re victims”. Empowering the educational system and uplifting Hawaiian values requires a shift of mindset both internally as well as throughout the settler community.

#### 3.8.2. Hoʻihoʻi ea (Re-Empower Community)

Many professional participants described the need for community re-empowerment at all levels. Lawelawe noted where empowerment was lacking, from the individual to parents and families to employees in the Department of Education who “are not empowered by the upper leadership”. After generations of loss and systemic racism, re-empowering the community to move together in advocacy and engagement was seen as a path towards healing historical trauma.

Several lawelawe saw that uplifting the voices of ʻōpio and other marginalized members of the community was an important component of community re-empowerment. One lawelawe felt that “more advocating for our ʻōpio that may not have voices” was needed to address historical trauma at a wider, policy level. One lawelawe who worked directly with ʻōpio advocates on policy changes shared that, “my young people and myself and our partners, have gone [to the legislature] three times and passed three bills”. This lawelawe emphasized that though they’ve done work at the legislative level, a key part of this advocacy was “practice implementation outside of the legislative forum” going on to say that “we don’t have to sit around and wait for someone else to change a bill or law. You know the healing and those things, it has to happen here. It has to happen in our community”.

In addition to amplifying ʻōpio voices, education on historical trauma was also viewed as an important step to unify and re-empower the community. One lawelawe shared that educating on history, especially historical trauma, might help to relieve the anger, fear, and blame some struggle with. This lawelawe shared that, “the fact that people are so angry and so fed up and they’re … they don’t know who to blame. So then they blame themselves. But if they were educated in the sense that historical trauma is a real thing that we’re all trying to battle and face on a daily basis and it is unjust and unfair. I think that would empower them to learn more and to be more inline with their history and their culture and use that as a tool to grow and help other people in similar situations”. In this sense, uniting the community through a deeper understanding of their shared history and trauma would offer re-empowerment and move towards healing and support together.

#### 3.8.3. Ka Hoʻoulu Alakaʻi (Grow Leaders)

NH ʻōpio expressed the belief that the current leaders were not up to the task. “We need to elect people who are proven to support the people, not corporate interest,” was how one participant articulated their views. In contrast, they felt that NH leaders of the past had the best intentions for all people, saying, “the leaders we had before, like our original founders were very akamai”. Another participant noted, “a lot of Hawaiian culture is to take care of other people too, so I think if we were still in those days, or not those days, but we still had a monarchy, more people would be taken care of”. One ʻōpio participant shared that “if we still had Native Hawaiians running our government, still leading us, I would have been put in safer foster homes” while another shared “even if we weren’t placed with our families, we would be placed with families that actually care. We’d be placed with families that would protect us”. Another ʻōpio shared that “if we had leadership in the Hawaiian community, that, and agreement, we could move forward”. NH ʻōpio, thus, felt that the current leaders today were unable to maintain “control” and the government was descending into “chaos”.

Several lawelawe discussed the need for more leaders who cared about NHs. Having “More Hawaiian leader[s]” was seen as one way to accomplish this. One participant noted that “the people that are really in charge, that have historically been in charge, number one—they don’t see things the same way that I do”. However, another lawelawe indicated that they felt that politicians needed to be educated about NH history and challenges. One participant shared, “I don’t think [politicians] know[] anything about cultural trauma”. One politician even shared that among other politicians there was a “lack of understanding” that community and academics needed to bridge, especially in relation to the continuing effects of historical trauma. Engaging current leadership by making them aware of the issue, providing evidence to support the importance of the issue, helping them understand how to attack it, and framing the importance of the issue within the current dialogue is something that politicians said they needed in order to tackle an issue.

#### 3.8.4. Ka Nohona ʻaeʻoia (Increasing Self-Sufficiency)

Native Hawaiian self-sufficiency was discussed as a source of pride for both ʻōpio and professional participants. ʻŌpio recognized how their ancestors used land to provide for themselves, “they used the land to, um, to feed themselves. Like it was very self-sustaining”. One provider reflected on the pride shared in self-sufficient practices like fishing and hunting in their own lifetime, “there was such a pride in fishing. And there was such a pride in hunting. And some, sometimes I hear [ ] the pride in hunting, in fishing, in farming”. Working with the land not only provides nourishment for Native Hawaiians, but also instills a sense of pride and connection.

Self-sufficiency was also discussed in terms of a desire for control. One ʻōpio commented that now, we have a lot of food shipped in”. Another shared that if Hawaiians still governed we would “probably [be] feeding ourselves instead of shopping at grocery stores”. Some professionals echoed this need for self-sufficient food production and pointed out some barriers that have prevented restoring these practices in the past, “we can’t find farmers. And it just came to me that the trauma of the farmers doing um, our pineapple and sugar industry when all the lunas [plantation supervisors] were White”.

ʻŌpio that engaged in hunting, fishing, or farming seemed more confident in their ability to survive. One ʻōpio shared that “if you could get that hunting skills down, that’d be good. Or fishing down too. You set”. One lawelawe shared that they had to shift their way of thinking when dealing with clients who were houseless and living on the beach. “They wanna live on, you know, the beaches. They’re self-sufficient. They don’t have to spend a lot of money”. Thus, in some instances being self-sufficient allows some NHs to feel like they have more control.

After sharing how they wished they could “get Hawaiʻi back,” one ʻōpio participant acknowledged there is still work to be done in restoring cultural practices in order to survive, especially with food and ʻāina. This ʻōpio shared that, “And without [cultural practices], how will we be able to take care of ourselves? Make sure we’re being fed, we’re getting something to drink? If we were to actually get everything back, and we were to be governed by our own people again, if we can’t grow our own crops, what good is that?” In their view, efforts to be as self-sufficient as before would be difficult without restoration of cultural practices. In fact, cultural practices were often connotated with survivance.

## 4. Discussion

NHs suffer from significant health and wellness disparities posited to stem from the harmful effects of colonization. Our findings indicate that NH ʻōpio experience historical trauma in a variety of ways, including through strong emotions that are difficult for ʻōpio to control; engaging in escapism; feeling ʻāina related harms; being caught up in messy systems; experiencing internal family, and community conflict; and feeling like certain things are not meant for them. NH ʻōpio also lean on the support of family, but may be disappointed by them or feel their absence. Finally, community and educational empowerment was seen as a way to mitigate harms linked historical trauma. Our findings align with other work that shows that while colonization has had a negative impact on Indigenous people, culture may mediate that impact [43,51,52].

### 4.1. Limitations

There were several notable limitations to this study. First, this study was conducted during the COVID-19 pandemic when travel was limited. As a result, all of the NH ʻōpio that participated were located on the island of Oʻahu. While Oʻahu is the most populated island, it is also the most urban and so the experiences of these ʻōpio are likely different from those on other islands. To minimize this limitation, we were able to schedule talk stories with lawelawe from other islands who worked closely with ʻōpio in those communities. However, to truly capture the variety of NH experiences participants from all islands would be ideal.

Second, the target age group 15–24 years old represented a large cognitive range. We focused on this age group because it aligned with the WHO definition of ʻōpio [53,54,55]. However, cognitive development does not necessarily match age in years, especially given the high rates of substance use from our participants and potential in utero substance use. Based on our observations we believe the ʻōpio talk stories represented a variety of developmental ages. As a result, not all ʻōpio participants were able to articulate the causes of their experiences to the same degree. Despite this many ʻōpio spoke of the same general experiences.

Third, throughout our talk stories, there appeared to be a divergence of cultural knowledge among ʻōpio participants. Some ʻōpio were able to articulate their understanding of Hawaiian history and how events such as the illegal overthrow of the Kingdom of Hawaiʻi or the banning of ʻōlelo Hawaiʻi impacted their families and their own lives. However, several ʻōpio participants did not always connect historical events to their lives or speak much of life before non-Hawaiians arrived in Hawaiʻi. There seemed to be three contributing factors to this divergence—education in a kaiapuni school, exposure to family stories, and age differences.

Kaiapuni schools or Hawaiian language immersion schools deliver instruction exclusively through ʻōlelo Hawaiʻi or Hawaiian language. ʻŌpio participants who reported attending kaiapuni or immersion programs were more likely to indicate knowledge of culture and history and connect the trauma of colonization to present-day problems in their community. These ʻōpio would also describe their feelings of extreme pride when engaging in cultural practices such as oli (chanting) at school. Considering the call for more Hawaiian-based education in many of our talk stories, more financial support for kaiapuni should be provided. Additionally, kaiapuni schools appear to be excellent models for the development and implementation of more culturally based education across Hawaiʻi.

The next contributor to a divergence in cultural knowledge is exposure to family stories. ʻŌpio were able to describe changes in Hawaiʻi based on the stories shared by older family members. Grandparents, parents, aunties, and uncles played key roles in bestowing cultural stories on ʻōpio, highlighting the importance of family in the transmission of culture. Taking into account this role of family, we can see that the separation of families or loss of family members to early death has devastating effects on overall cultural knowledge. Without older relatives to guide and educate ʻōpio, future generations will likely continue to suffer a disconnect from their culture resulting in further loss for NHs.

The final factor affecting the divergence in cultural knowledge among ʻōpio was age. As expected, older ʻōpio were more likely to accurately describe Hawaiian history or connect the events from the past with current circumstances. Older ʻōpio were more likely to recognize that the detrimental patterns in their family history stemmed from past trauma and loss. Further, ʻōpio who had both experience/connectedness with family stories and in kaiapuni schools may be best positioned to articulate Native Hawaiian historical trauma and its effect on themselves and their community. These patterns lend evidence to historical trauma’s manifestation throughout the lifespan. As individuals mature and develop critical thinking skills, awareness of their position in a broader historical context becomes clearer. These developments may leave NHs more vulnerable to symptoms like depression if not addressed.

### 4.2. Policy Recommendations

Because historical trauma was brought up by so many lawelawe in a variety of position, including in the DOE school system, the authors were able to collect and synthesize several policy recommendations. First, historical trauma has not been acknowledged by the state of Hawaiʻi. While some laws would recognize the detrimental impacts of colonization and racism in Hawaiʻi as well as health disparities among NHs, there was never a clear acknowledgment that colonization has continuing impacts on NHs. To address this gap, we recommend that the Hawaiʻi Legislature acknowledge NH historical trauma as an issue. The authors understand that this would require the state to accept a certain amount of responsibility, however, identifying and prioritizing historical trauma is the first step towards community healing. Related to acknowledgement, to better understand NH historical trauma and how it relates to negative outcomes (i.e., health disparities, incarceration rates, etc.), we recommend that the Hawaiʻi Legislature establish a task force to study NH historical trauma and its impact.

Additionally, while bills that incorporated NH restorative processes or healing were often introduced and referred to committee, they did not pass the legislature to become law. Culturally based programs, especially ʻāina-based programs, have produce positive effects among NH ʻōpio [56,57,58]. To effectively address the issue of historical trauma, we recommend that the Legislature support policies or programs that center NH restorative processes or offer culturally based healing for NH ʻōpio and families. In particular programs that are integrated into the general curriculum at elementary, middle, and high schools have the potential to reach a broad student population providing healing opportunities as well as opportunities to increase cross-cultural understanding and respect.

### 4.3. Future Studies

Historical trauma was called out as one of the priority focus areas of NH health in the 2020 Assessment of Native Hawaiian Priorities for Health and Wellbeing [59]. Native Hawaiians have recognized historical trauma as a critical area for policymakers to address. Yet, politicians and decision makers appear to be more skeptical about the need to tackle historical trauma. Our legislative contacts clearly indicated that in order to work on an issue legislators need to be aware of the issue and need evidence-based support for its importance. The lack of a scale to measure NH historical trauma, thus, is a barrier to moving forward with a policy solution as it impacts problem identification [60]. Creating a scale to measure NH historical trauma has the potential to provide the evidence that policymakers need to tackle historical trauma.

Because historical trauma impacts individuals differently based upon their experiences as well as their built up resilience, it is prudent to look at NH historical trauma experiences across the lifespan. We know from work in other communities that experiences are not fixed, but rather are interpreted and re-interpreted based on age and circumstance [61]. We also have some evidence to indicate that cultural engagement and knowledge mediates against historical trauma. Thus, while there is a need to develop a NH focused measure for historical trauma, it is important to consider this concept from a variety of NH perspectives. Future studies should explore NH experiences with historical trauma across the lifespan, across the paiʻāina, and across various cultural competencies before developing a mechanism to measure NH historical trauma.

Finally, it is insufficient to merely catalog current and ancestral trauma. To improve Indigenous wellbeing, we must begin to focus on healing the cultural and current trauma that so many in the NH experience. Educators should be trained on NH history and be comfortable addressing these difficult topics in the classroom. Curriculum should be developed that incorporates Hawaiian cultural practices and knowledge, building pride and fluency for all ʻōpio, including non-Hawaiians. More work is needed to understand how best to heal NH historical trauma, however, it will likely require a multiplicity of collective efforts since it was created through a multiplicity of actions over time.

## 5. Conclusions

Historical trauma has been linked with several physical and mental health disparities, including substance use among American Indians. Less work has been done on historical trauma among Native Hawaiians, who experienced similar but unique forms of colonization. Our findings based on 34 talk story sessions with NH ʻōpio and lawelawe indicate that NH ʻōpio experience historical trauma in a multitude of ways that have very real impacts on their present day lives. Participants articulated eight themes covering individual, community, and systemic domains, including: (1) Peʻa ka lima i ke kua (Emotions); (2) ʻAuana i ke kula ʻo Kaupeʻa (Escaping from grief); (3) He aliʻi ka ʻāina (Land is a chief); (4) Kū i ka welo (Family Connectedness); (5) Kūneki nā kūʻauhau liʻiliʻi, noho i lalo (Dreams, but not for me); (6) Ke aupuni ʻānunu o Halaea (Messy System); (7) Hihia Hīkākā (Entangled); and (8) Hoʻi i ka pono (Restoration of Balance).

Despite the significant impacts that historical trauma has on NH ʻōpio, many expressed pride in their identity and provided multiple hopeful statements about their future. The authors are, thus, inspired by an ʻōlelo noʻeau (proverb/wise saying) gifted to us by our ancestors: He pūkoʻa kani ʻāina translated as a coral reef grows into an island. This ʻōlelo noʻeau signifies the ability of something small to grow over time into a sustainable, productive, and nurturing being. NH ʻōpio have the potential to grow into that island with the support of the community and trauma-informed policies that incorporate healing historical trauma.

## Figures and Tables

**Table 1 ijerph-19-12564-t001:** Demographics (*n* = 34).

Participants	Organizational Affiliations
ʻŌpio (youth) (*n* = 19) Minors (*n* = 12) Male (*n* = 6) Female (*n* = 6) 18–24 (*n* = 7) Male (*n* = 2) Female (*n* = 5)	Adult Friends for Youth (East Oʻahu and West Oʻahu) Residential Youth Services & Empowerment (Windward Oʻahu) Kawailoa Youth and Family Center—Hawaiʻi Youth Correctional Facility (Windward Oʻahu)
Lawelawe (Service Providers) (*n* = 15) Male (*n* = 3) Female (*n* = 12) Native Hawaiian (*n* = 9)	2 School Psychologists & Counselors (West Oʻahu, Windward Oʻahu) 2 Correctional Facility Staff (Windward, Oʻahu) 2 Hawaiʻi State Legislators (across Oʻahu) 2 Child Welfare Service Investigators (across Oʻahu) 2 Judiciary & Juvenile Probation (across Oʻahu) 5 Social Workers & Advocates (Oʻahu, Maui, and Hawaiʻi Island)

**Table 2 ijerph-19-12564-t002:** Themes.

Themes	Sub-Themes
Theme 1: Peʻa ka lima i ke kua (Emotions)	Ke kaumaha (Pain/Sadness)
	Ka huhū (Anger)
	Mānewanewa (Loss of Control)
	Pau ka pono (Hopelessness)
	Ka hopohopoalulu (Anxiety/Fear)
	Ka haʻaheo (Pride)
Theme 2: ʻAuana i ke kula ʻo Kaupeʻa (Escaping from grief)	ʻAi lāʻau ʻino (Substance Use)
	Lilo i ka ʻenehana (Technology as a Distraction)
	Ka hōʻalo (Avoidance)
	ʻImi Loaʻa (Consumerism)
Theme 3: He aliʻi ka ʻāina (Land is a chief)	Ka lilo ka ʻāina (Land Loss)
	Ka hana ʻino i ka ʻāina (Environmental Destruction)
	Ka hoʻihoʻi ʻāina (Return Land)
	Ke aloha ʻāina (Land as Healing)
Theme 4: Kū i ka welo (Family Connectedness)	Ka ʻohana (Family Support System)
	Ka hilihewa o ka makua i ke ʻano haole (Distortion of Parental Values)
	Ke kauhale (Connectedness to Extended ʻOhana and Community)
	Ke Kupuna (elders)
Theme 5: Hoʻohikihiki (Dreams, but not for me)	Ka nohona Hawaiʻi kahiko (Thriving in the Past)
	Ka waiwai kālā (Money/Haves and Have Nots)
	Nā hale (Housing)
	Ka hōʻole kahuna (Anti-intellectualism)
Theme 6: ʻŌnaehana Huikaulua (Messy System)	Ka pono waiwai haole (Colonialism and Other -isms)
	Ka malihini (Tourism)
	Ka pūʻali koa (Militarism)
	Ka ʻehaʻeha ma ke kula (School Re-traumatizes)
	Nā kōkua kamaliʻi (Child Welfare Services)
	Ke olakino kauʻwa (Health Disparities/COVID)
	Ka hoʻopaʻa hao (Carceral System)
Theme 7: Hihia Hīkākā (Entangled/Sprawling Trouble)	Nā pilikia kaiaulu (Community Conflict)
	Nā pilikia kūloko (Internal Conflict)
Theme 8: Hoʻi ka pono (Restoring Balance)	Ka hoʻomana (Empowered Education)
	Hoʻihoʻi ea (Re-Empower Community)
	Ka hoʻoulu alakaʻi (Growing Leaders)
	Ka nohona ʻaeʻoia (Increase Self-Sufficiency)

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
