# Peer review of "Ke ala i ka Mauliola: Native Hawaiian Youth Experiences with Historical Trauma"

_ijerph, 2022, doi:10.3390/ijerph191912564_

Round 1
Reviewer 1 Report
This study addressed the gap in the literature by conducting a qualitative analysis of Indigenous talk stories with Native Hawaiian opio and ka lawelawe. The results indicate that if left unaddressed, a significant part of the future generation will be at risk of failing to reach their full potential.
Very interesting article and timely given that the past impacts the future and the current issues facing Indigenous populations around trauma and its connection to the disruption from colonization and assimilation.
The reference at 15 needs to be identified. I wasn't sure if it was an article or a chapter in a book but when I checked the title and author, I see that it is an electronic poster presentation.
Author Response
We greatly appreciated Reviewer 1’s comments and detail oriented review of our citations. We are pleased that the Reviewer felt that this manuscript addressed a gap in the literature.
- The reference at 15 needs to be identified. I wasn't sure if it was an article or a chapter in a book but when I checked the title and author, I see that it is an electronic poster presentation. There are two places in the paper where claims are made but not substantiated by citations.
Response: We have corrected the citation for referencer 15, including adding additional information (i.e., link) to citation to aid in viewing. We have also re-reviewed the manuscript to ensure that all claims are substantiated by citations adding citations in several places.
Reviewer 2 Report
The paper is well written and researched. The methods are appropriate, the topic is important and the findings connect to their policy recommendations. My only critique is that the results section could possibly be integrated in a different way. As of now, it is broken into sections based on themes with quotes that highlight the theme. It is interesting and works but it reads more like a list. This could be just a personal preference. This is a minor critique of an otherwise well written paper.
Author Response
We appreciate the enthusiasm of Reviewer 2’s comments and are gratified that the manuscript spoke to the Reviewer. Moreover, we appreciate the reminder to try to add interest to our writing style.
- My only critique is that the results section could possibly be integrated in a different way. As of now, it is broken into sections based on themes with quotes that highlight the theme. It is interesting and works but it reads more like a list. This could be just a personal preference. This is a minor critique of an otherwise well written paper.
Response: We appreciate Reviewer 2's comment on the integration of the results section. We spent some time thinking through alternative formats. While we agree that the results currently reads like a list, we came to the conclusion that this format is the most logical way to share the results. We are, however, open to concrete alternatives, if suggested.
Reviewer 3 Report
Fantastic article, I thoroughly enjoyed reading the article and feel that it would be a crucial addition to the work on historical loss. The introduction was well organized and succinct. Thank you for your important work!
Methods:
- I suggest adding a bit more information on the demographics, particularly given that the recruitment was not random, but from organizations. What is the gender breakdown, SES, and any other variables that are available to disclose?
Results:
- Not required, but consider including the concept of cultural bereavement (see work of Bhugra & Eisenbruch) when you discuss feelings towards cultural loss in the "Internal conflict" section.
- line 1162, the section titled "Uplift Leaders" is a bit confusing, maybe reword to "Need for Trust in Leaders"
Formatting:
- keep capitalization consistent. For example, line 562 Connectedness should be capitalized. Line 698 should have a comma.
- line 808, there should be a "a" before messy and messy should either not be capitalized or all the themes should be capitalized.
- consider cleaning up the sentence starting on line 862 by splitting up the two separate points before and after the "when." it might be helpful to further, explain what the participant meant by the moratorium.
- Line 871: consider a stronger introductory sentence to this section. For example, it can start with "Participants often brought up the connection between military work and their feelings of historical loss."
- the sentence in line 899 is confusing. Is it supposed to say, "Teachers that did not demonstrate care for their students..."
- Line 907: consider using less passive voice. FOr example, the sentence. "One of the challenges that was frequently brought up was that a lot of teachers are brought in “from the mainland." could instead be written as "many of the participants reflected on the impact of the lack of NH representation in their teachers."
line 1236, there needs to be a comma after "to minimize this limitation"
Author Response
We greatly appreciated the level of detail that Reviewer 3 provided. We believe that we can always improve on our work and appreciate the efforts of Reviewer 3. We have no doubt that the quality of this paper was improved as a result of addressing Reviewer 3’s comments.
- Methods: I suggest adding a bit more information on the demographics, particularly given that the recruitment was not random, but from organizations. What is the gender breakdown, SES, and any other variables that are available to disclose?
Response: We agree with Reviewer 3 that more detail would be useful. To respond to this comment, we added the gender breakdown and clarified that all of the youth are considered at-risk. We did not ask any additional demographic questions as we were more concerned about whether they met the criteria of juvenile incarceration and then the criteria set out in the Historical Loss Scale, which did not list SES as a factor in historical trauma. In addition, we wanted to ensure anonymity and therefore opted to collect less demographic data. For the lawelawe we added gender and whether or not they were Native Hawaiian. Finally, we categorized participants by professions to highlight the diversity of our sample. We did not collect any additional demographic data.
- Results: Not required, but consider including the concept of cultural bereavement (see work of Bhugra & Eisenbruch) when you discuss feelings towards cultural loss in the "Internal conflict" section.
Response: We reviewed the article provided and found a significant number of similarities between cultural bereavement and some of our findings, especially within the internal conflict section. However, the concept and article are grounded in status as refugees who have been forced out of their home countries. In the Native Hawaiian context we remain in their homelands, but have had our culture forcibly removed/taken. Thus, we believe that the similarities in these populations surrounding the ideas of cultural bereavement and cultural loss warrant a deeper exploration that we can provide in this paper.
- Results: line 1162, the section titled "Uplift Leaders" is a bit confusing, maybe reword to "Need for Trust in Leaders"
Response: We have reconsidered our terminology and have adopted the term “Growing Leaders” over Need for Trust in Leaders as it encompasses more than just trust and it articulates the active role that some participants envisioned for the community.
- Formatting: keep capitalization consistent. For example, line 562 Connectedness should be capitalized. Line 698 should have a comma.
Response: Mahalo for catching the capitalization inconsistency. We have reviewed all capitalization of terminology in the manuscript. We also added the comma that the reviewer mentioned.
- Formatting: line 808, there should be a "a" before messy and messy should either not be capitalized or all the themes should be capitalized.
Response: Mahalo for pointing this out. We have made “messy/confused” lower-case. We intend for the themes to be capitalized in the heading, but not in the text. As such, we have gone back and fixed a few additional themes. However, we believe that the referenced sentence is grammatically correct since we are referencing the title of the theme.
- Formatting: consider cleaning up the sentence starting on line 862 by splitting up the two separate points before and after the "when." it might be helpful to further, explain what the participant meant by the moratorium.
Response: We revised this section again. In one place splitting a sentence in order to increase readability and in other places reworking the sentence for better flow. We also added in a sentence explaining that the moratorium referred to a proposed policy that would limit the ability of individuals (especially non-residents) from purchasing housing in Hawaiʻi for the purpose of short-term rental thereby removing the housing unit from the residential market.
- Formatting: Line 871: consider a stronger introductory sentence to this section. For example, it can start with "Participants often brought up the connection between military work and their feelings of historical loss."
Response: Mahalo for this recommendation. We have accepted it and revised our manuscript accordingly.
- Formatting: the sentence in line 899 is confusing. Is it supposed to say, "Teachers that did not demonstrate care for their students..."
Response: We have reviewed the follow sentence and have added clarifying language.
- Formatting: Line 907: consider using less passive voice. For example, the sentence. "One of the challenges that was frequently brought up was that a lot of teachers are brought in “from the mainland." could instead be written as "many of the participants reflected on the impact of the lack of NH representation in their teachers."
Response: We appreciate this suggested and have activated this sentence in our revision.
- Formatting: line 1236, there needs to be a comma after "to minimize this limitation"
Response: Mahalo for catching this. We have added the comma in the revision.